# Skeletal reorganization divergence of *N*-sulfonyl ynamides

Linwei Zeng[1], Yuxin Lin [1], Jiaming Li[1], Hironao Sajiki [2], Hujun Xie[3] & Sunliang Cui [1✉]

Skeletal reorganization is a type of intriguing processes because of their interesting mechanism, high atom-economy and synthetic versatility. Herein, we describe an unusual, divergent skeletal reorganization of *N*-sulfonyl ynamides. Upon treatment with lithium diisopropylamine (LDA), *N*-sulfonyl ynamides undergo a skeletal reorganization to deliver thiete sulfones, while the additional use of 1,3-dimethyl-tetrahydropyrimidin-2(1*H*)-one (DMPU) shifts the process to furnish propargyl sulfonamides. This skeletal reorganization divergence features broad substrate scope and scalability. Mechanistically, experimental and computational studies reveal that these processes may initiate from a lithiation/4-exo-dig cyclization cascade, and the following ligand-dependent 1,3-sulfonyl migration or β-elimination would control the chemodivergence. This protocol additionally provides a facile access to a variety of privileged molecules from easily accessible ynamides.

[1] Institute of Drug Discovery and Design, College of Pharmaceutical Sciences, Zhejiang University, 866 Yuhangtang Road, 310058 Hangzhou, China. [2] Laboratory of Organic Chemistry, Gifu Pharmaceutical University, Gifu 501-1196, Japan. [3] School of Food Science and Biotechnology, Zhejiang Gongshang University, 310018 Hangzhou, China. ✉email: slcui@zju.edu.cn

Skeletal reorganization is a type of process involving multiple bond cleavage and formation for molecule framework reassembly (Fig. 1a)[1,2]. Owing to the intriguing reaction mechanism, high atom-economy property and capability of accessing complex and synthetically challenging molecules, the skeletal reorganization process has attracted considerable attention and also been widely applied in organic synthesis toward diverse carbocyclic and heterocyclic compounds[3–5]. For example, a variety of transition-metal-catalysed skeletal reorganizations of enynes have been explored for efficient synthesis of polycyclic compounds[6–10]. Recently, Sun and co-workers discovered an unusual skeletal reorganization of oxetane for the synthesis of 1,2-dihydroquinolines (Fig. 1b)[11]. Meanwhile, Liu and co-workers established a skeletal reorganization protocol of olefine via a radical-initiated cyclization/1,n (n = 3, 4, 5) vinyl migration cascade for accessing medium- and large-sized cycles, which are ubiquitous structural motifs in natural products and pharmaceutical agents (Fig. 1c)[12]. Very recently, Zhu and co-workers reported a novel skeletal reorganization of kojic acid- or maltol-derived alkynes under Indium catalysis, which provided an expeditious access to valuable hydroxylated benzofurans (Fig. 1d)[13]. Despite these advances, the investigation of skeletal reorganizations toward synthetically challenging and biologically interesting molecules remains continuously interesting and important.

The four-membered sulfur-containing heterocycles, such as thiete sulfones, thietane sulfones and thietanes, are strained small ring compounds that have found wide applications in the discovery of dye, drug and pesticide (Fig. 2)[14–19]. Typically, the unsaturated thiete sulfones showed valuable synthetic utility in organic synthesis. For example, thiete sulfones could be used as dienophiles in the Diels–Alder reaction with tetraphenyl cyclopentadienones or isobenzofurans for accessing bridged and fused-ring compounds[20–22]. Moreover, they could participate in [3 + 2] cycloadditions with diazo compounds or nitrile oxides for the synthesis of heterocycles[23,24]. Recently, thiete sulfones have been investigated in C–H functionalization to establish axially chiral molecules and macrocyclic compounds[25,26]. However, the conventional synthesis of thiete sulfones relies on multi-step routes and also suffers from narrow scope[27–32]. Therefore, it would be interesting to explore a distinct and efficient approach to functionalized thiete sulfones.

Ynamides are a type of N-substituted electron-rich alkynes that exhibit unique chemical properties and serve as versatile synthons in organic synthesis[33–42]. For example, ynamides could act as flexible cyclization partners in heterocycle synthesis[43–45], carbene precursors[46–49] and enamide precursors[50,51], racemization-free coupling reagents for peptide and macrolide synthesis[52–54] and C2 building blocks of multicomponent reactions)[55–57]. In recent years, the intramolecular cyclizations of ynamides, including transition-metal-catalysed and Brønsted acid-catalysed nucleophilic cyclizations, anionic cyclizations and radical cyclizations, have been extensively investigated for the synthesis of N-heterocycles (Fig. 3a)[58–61]. However, the skeletal reorganization of ynamides is rarely reported. In 2012, Evano and co-workers reported an s-BuLi-mediated skeletal reorganization of N-Boc ynamides for de novo synthesis of 1,4-dihydropyridines and pyridines, which invoked a process of carbonyl-directed deprotonation and anionic 6-endo-dig cyclization (Fig. 3b)[62]. Encouraged by these, we hypothesized that the deprotonation at the α-position of sulfonyl moiety of N-sulfonyl ynamides might initiate an anionic 4-exo-dig or 5-endo-dig cyclization to deliver cyclic sulfonamides, which are privileged structures in medicinal chemistry (Fig. 3c)[63,64]. In continuation of our interests in ynamide chemistry[55–57,65], herein we would like to report a skeletal reorganization divergence of N-sulfonyl ynamides for selective entry to thiete sulfones and propargyl sulfonamides (Fig. 3d).

## Results

**Reaction optimization.** We commenced our study by using N-sulfonyl ynamide **1a** and lithium base to investigate this reaction.

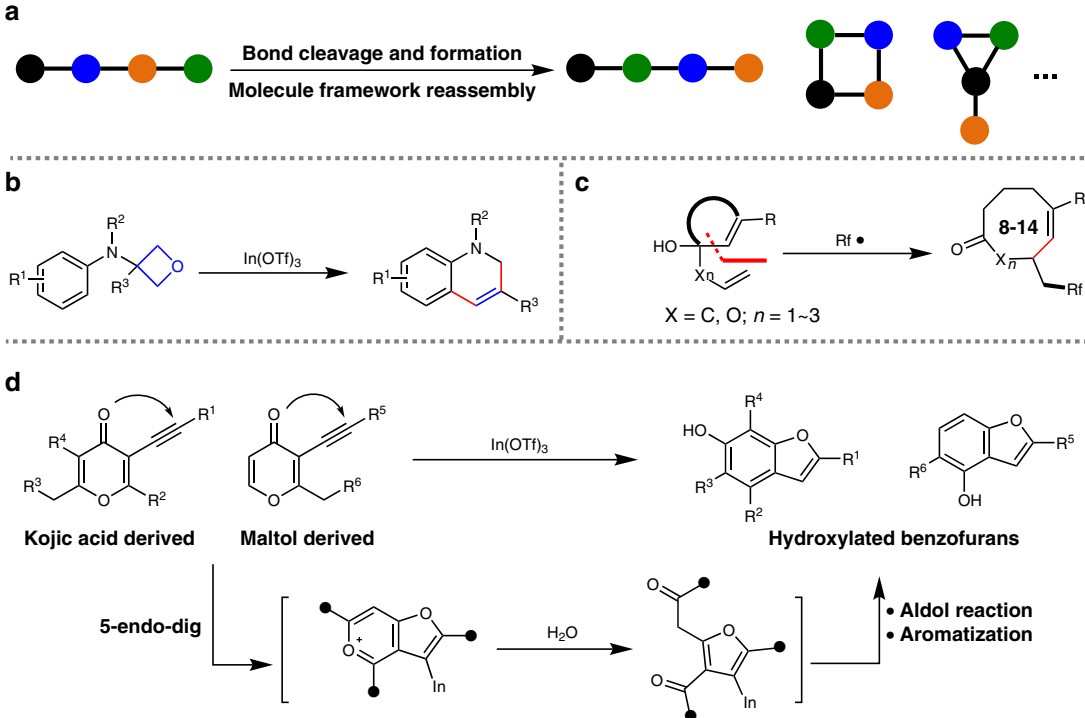

**Fig. 1 Skeletal reorganization. a** Scheme of a skeletal reorganization in molecules. **b** In-catalysed skeletal reorganization of oxetanes. **c** Radical-initiated skeletal reorganization of olefines. **d** In-catalysed skeletal reorganization of kojic acid- or maltol-derived alkynes.

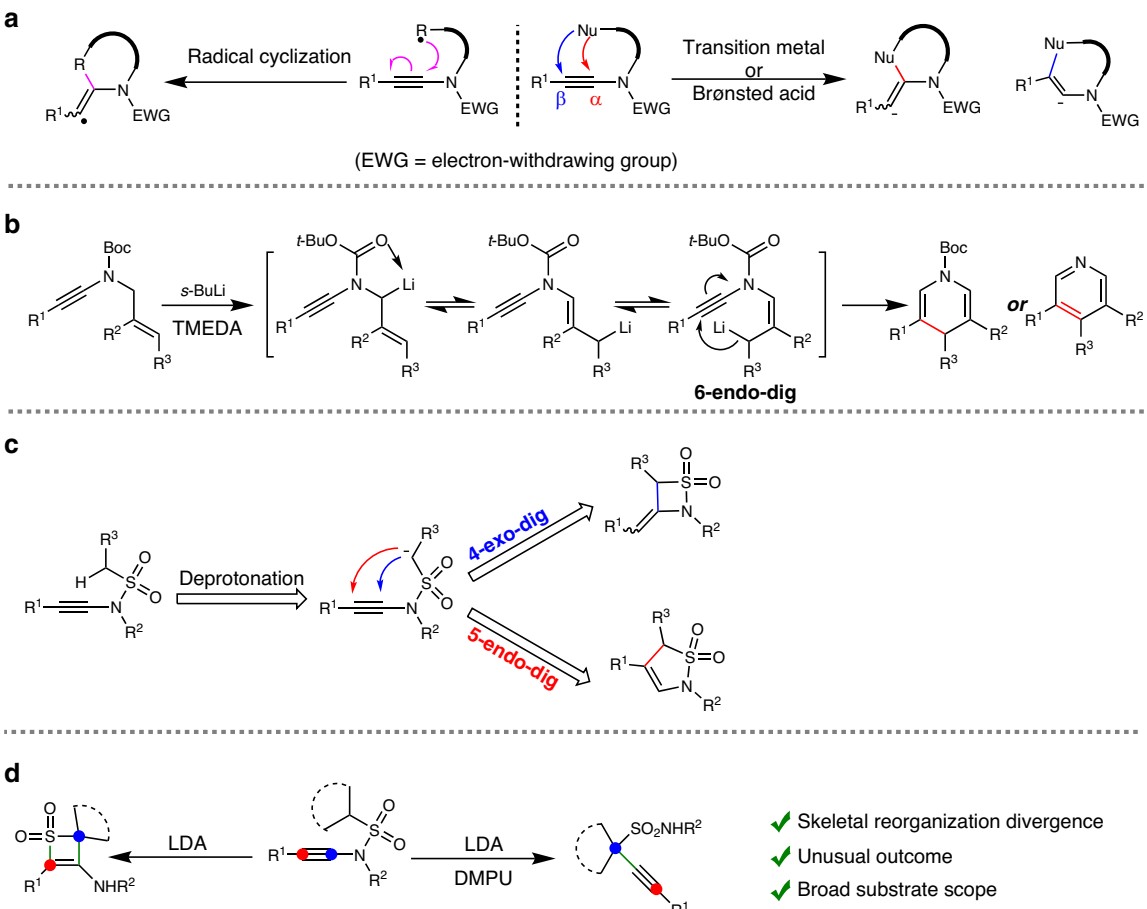

**Fig. 2 Sulfur-containing four-membered ring in useful molecules.** Some representative cases are listed.

**Fig. 3 Skeletal reorganization of ynamides. a** Intramolecular cyclizations of ynamides. **b** Skeletal reorganization of *N*-Boc ynamides. **c** Our initial hypothesis. **d** This work: Skeletal reorganization divergence of *N*-sulfonyl ynamides.

As shown in Table 1, we initially treated **1a** with *n*-BuLi at −40 °C for 1 h and then quenched with MeOH; a new major product **2a** and a minor product **3a** were isolated (entry 1). The standard analysis, including ¹H nuclear magnetic resonance (NMR), ¹³C NMR and mass spectroscopy, was not able to identify these compounds. Gratifyingly, the X-ray analysis showed that **2a** was a thiete sulfone and **3a** was a propargyl sulfonamide (for more details, see Supplementary Figs. 8 and 9), indicating an unusual

occurrence of skeletal reorganization. This unexpected outcome prompted us to optimize the reaction. The next survey of lithium bases showed that lithium diisopropylamine (LDA) was superior to give **2a** in 72% yield (entry 2), while the utilization of lithium bis(trimethylsilyl)amide would decrease the yield to 31% (entry 3). The use of sodium bis(trimethylsilyl)amide and potassium bis (trimethylsilyl)amide as base could only give **3a** in very low yield (entries 4–5). Using NaH, 1,8-diazabicyclo[5.4.0]undec-7-ene or

**Table 1 Reaction optimization[a].**

| Entry | Base | Additive | Solvent | $T$ (°C) | Yield of 2a (%)[b] | Yield of 3a (%)[b] |
|---|---|---|---|---|---|---|
| 1 | $n$-BuLi | None | THF | −40 | 60 | 19 |
| 2 | LDA | None | THF | −40 | 72 | 18 |
| 3 | LiHMDS | None | THF | −40 | 31 | 0 |
| 4 | NaHMDS | None | THF | −40 | 0 | 14 |
| 5 | KHMDS | None | THF | −40 | 0 | 16 |
| 6 | NaH | None | THF | 25 | 0 | 0 |
| 7 | DBU | None | THF | 25 | 0 | 0 |
| 8 | TEA | None | THF | 25 | 0 | 0 |
| 9 | LDA | None | THF | 0 | 68 | 19 |
| 10 | LDA | None | THF | −78 | 65 | 21 |
| 11 | LDA | TMEDA | THF | −40 | 63 | 16 |
| 12 | LDA | DMPU | THF | −40 | Trace | 82 |

*LDA* lithium diisopropylamine, *LiHMDS* lithium bis(trimethylsilyl)amide, *NaHMDS* sodium bis(trimethylsilyl)amide, *KHMDS* potassium bis(trimethylsilyl)amide, *DBU* 1,8-diazabicyclo[5.4.0]undec-7-ene, *TEA* trimethylamine, *TMEDA* N,N,N',N'-tetramethylethylenediamine, *DMPU* 1,3-dimethyl-tetrahydropyrimidin-2(1H)-one.
[a]Reaction conditions: **1a** (0.2 mmol), base (0.3 mmol), additive (1 mmol), THF (2 mL), 1 h, then MeOH (0.1 mL).
[b]Isolated yields.

trimethylamine as base showed inferior to shut down the reactivity even at room temperature, and **1a** was recovered (entries 6–8). When LDA was used as base and the temperature was varied to 0 °C or −78 °C, the yield of **2a** would slightly decrease (entries 9–10). Considering that lithiation was involved in this process, we tried to add ligands to increase the yield. Next, using of N,N,N',N'-tetramethylethylenediamine as ligand showed no improvement (entry 11). Interestingly, the utilization of 1,3-dimethyl-tetra-hydropyrimidin-2(1H)-one (DMPU) as ligand would exclusively give **3a** in good yield (entry 12, 82% yield). Therefore, the skeletal reorganization is divergent and could be controlled by ligand.

**Reaction scope study.** With the optimized reaction conditions in hand, we next tested the substrate scope of this skeletal reorganization divergence. The starting material N-sulfonyl ynamides could be easily prepared by coupling of sulfonamides and alkynyl bromides. As shown in Fig. 4, various ynamides **1** could participate well in this skeletal reorganization, leading to the corresponding thiete sulfones **2** in moderate-to-good yields. Diverse substitutions on the amino group of ynamides, including n-butyl, benzyl, i-propyl, cyclopentanyl and thiophene-2-ethyl, were found tolerable in this process (**2b**–**2d**). The (S)-1-phenylethyl amine-derived ynamide **1g** could also engage in this reaction to give the chiral moiety tethered thiete sulfone **2g** in 77% yield. Other functional groups, such as alkene and protected alcohol, were also compatible in this reorganization process to give the corresponding products (**2h** and **2i**), which might offer ample opportunities for the further derivatization. In addition, the aniline-derived ynamide (**1j**) was also applicable to deliver the desired product **2j** in an excellent 92% yield. The structure of **2j** was further confirmed by X-ray diffraction (for more details, see Supplementary Fig. 10). For the variation of sulfonyl groups, a variety of substitutions were also tested and found amenable in this process to access functionalized thiete sulfones (**2k**–**2q**). In

detail, N-methylsulfonyl and N-benzylsulfonyl ynamides could reorganize to corresponding thiete sulfones in moderate yield (**2k**–**2m**), and the structure of **2k** was confirmed by X-ray diffraction (for more details, see Supplementary Fig. 11). When N-cycloalkylsulfonyl ynamides like N-cyclopentanylsulfonyl ynamides and N-cyclohexanylsulfonyl ynamides were used, the skeletal reorganization could deliver *spiro*-fused thiete sulfones in moderate-to-good yields (**2n**–**2q**), and the structure of **2q** was also verified by X-ray diffraction (for more details, see Supplementary Fig. 12). With respect of the substitution at the β-position of ynamides, a set of aryl groups functionalized with 2-chloro, 3-methyl, 4-methyl, 4-pentyl, 4-methoxy, 4-fluoro or 4-chloro were tolerable to furnish the corresponding products in good yields (**2r**–**2x**), and these substituents did not show any significant electronic and steric effects with the yields. Other aryl groups, including 2-naphthyl, 3-pyridinyl and thiophene-2-yl, were suitable in this process to produce the products successfully (**2y**–**2a'**), and the structure of **2a'** was confirmed by X-ray analysis (for more details, see Supplementary Fig. 13). Meanwhile, the cyclohexenyl-substituted ynamide **1d'** could smoothly transform to the corresponding product **2b'** in 62% yield. Notably, silyl substitution was compatible with the process as well. For example, the TIPS-substituted ynamide **1e'** could reorganize to product **2c'** in 58% yield with retention of the TIPS group, while the TMS-substituted ynamide **1f'** could transform to product **2d'** in a remarkable 94% yield accompanied with TMS desilylation. Unfortunately, β-alkyl-substituted and terminal ynamides were not applicable in this process (**2e'**–**2f'**). Since the amino-containing full-substituted thiete sulfones are synthetically challenging[66], this skeletal reorganization of ynamides provides a robust and efficient approach toward these molecules with the achievement of structural diversity and molecule complexity.

Next, the scope for another skeletal reorganization toward propargyl sulfonamide was also investigated. As shown in Fig. 5, a variety of ynamides were treated with LDA and DMPU toward

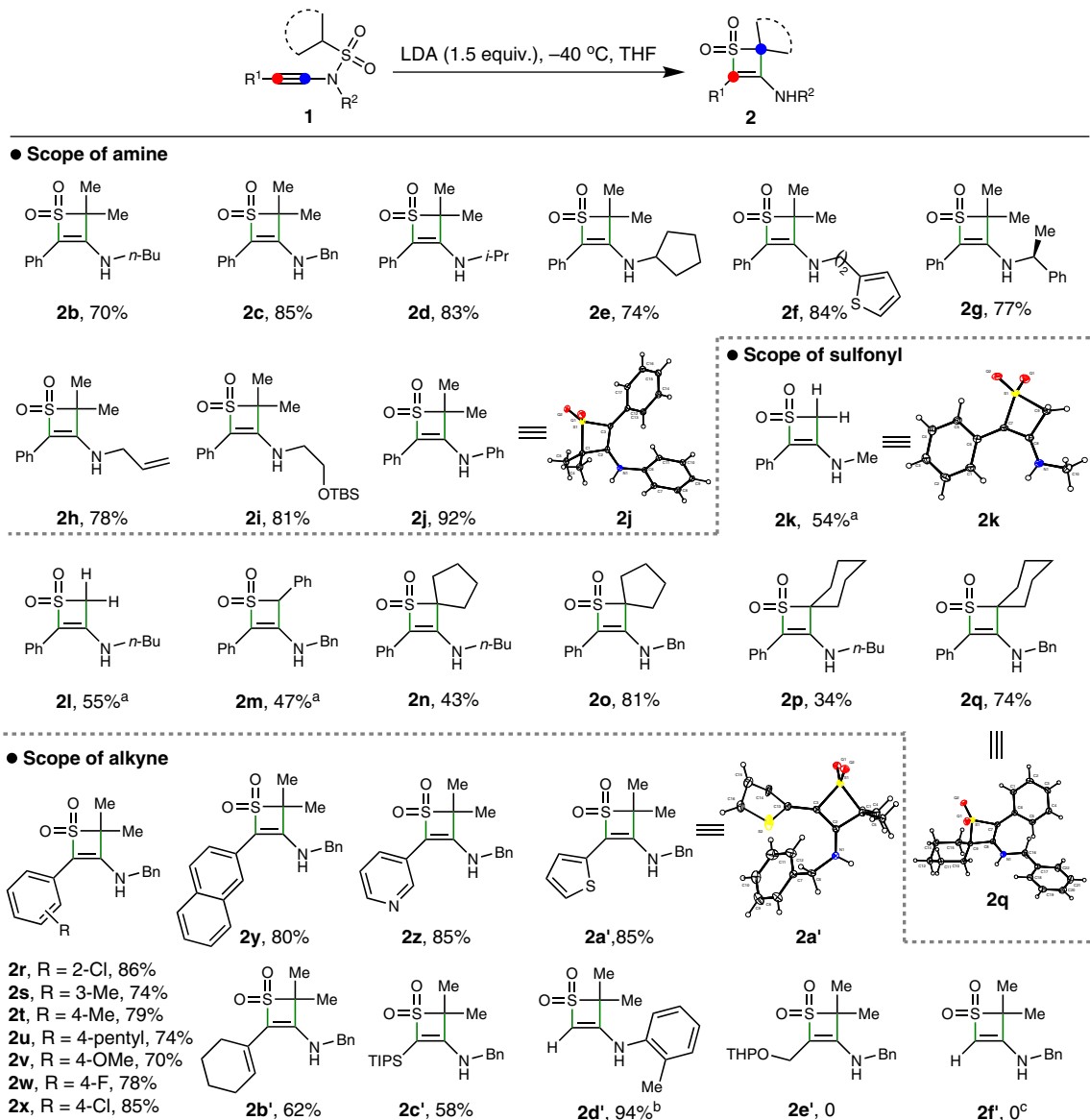

**Fig. 4 Scope of thiete sulfones.** Standard condition: **1** (0.2 mmol), LDA (0.3 mmol), THF (2 mL), −40 °C, 1 h, then treated with MeOH (0.1 mL). Yields refer to isolated products. [a]0.6 mmol LDA was used. [b]Product of desilylation. [c]From the terminal ynamide.

the formation of propargyl sulfonamides, and the amino substitutions like *n*-butyl, cyclopentanyl, thiophene-3-ethyl and protected alcohol were compatible with this process (**3b–3e**). With respect to the sulfonyl substitutions, the cyclopentanyl and cyclohexanyl were applicable in the process to deliver the corresponding products in good yields (**3f–3i**). Meanwhile, alkynes with substitution on the aryl ring, including 2-fluoro, 2-chloro, 3-methyl, 4-methyl, 4-pentyl and 4-fluoro, could well engage in this skeletal reorganization process to give the corresponding propargyl sulfonamides in good yields (**3j–3o**). The naphthyl and pyridinyl groups were also applicable to furnish the products smoothly (**3p–3q**). In terms of the structure of products, this skeletal reorganization involves an interesting 1,3-alkyne migration from *N*-atom to *C*-atom. Propargyl sulfonamide is a versatile synthon and their synthesis is challenging by conventional methods. Thus this protocol provides a simple and efficient method to synthesize these molecules from readily available materials along with the fascinating process.

**Synthetic applications and mechanism study**. To demonstrate the synthetic utility of this skeletal reorganization divergence, an 8 mmol scale reaction was conducted (Fig. 6a). Under the standard reaction conditions, **1a** could be selectively transformed to **2a** and **3a** in gram-scale, indicating that these skeletal reorganization processes were practical. Considering that these processes may involve lithium intermediates that could offer opportunities for divergent functionalization of products, the derivatization was carried out. As shown in Fig. 6b, when various electrophilic reagents instead of MeOH were used to quench the reaction after treating ynamide **1** with LDA, corresponding electrophilic groups, such as methyl, allyl, propargyl, protected ethanol-2-yl, acetate and acetyl, were successfully installed in the amino group of products, leading to a series of functionalized thiete sulfones (**4a–4f**). This result not only confirmed that a lithium amino intermediate is involved in the reorganization process but also offered a vast potential for further derivatization of thiete sulfone skeletons. Similarly, when the LDA/DMPU-mediated skeletal

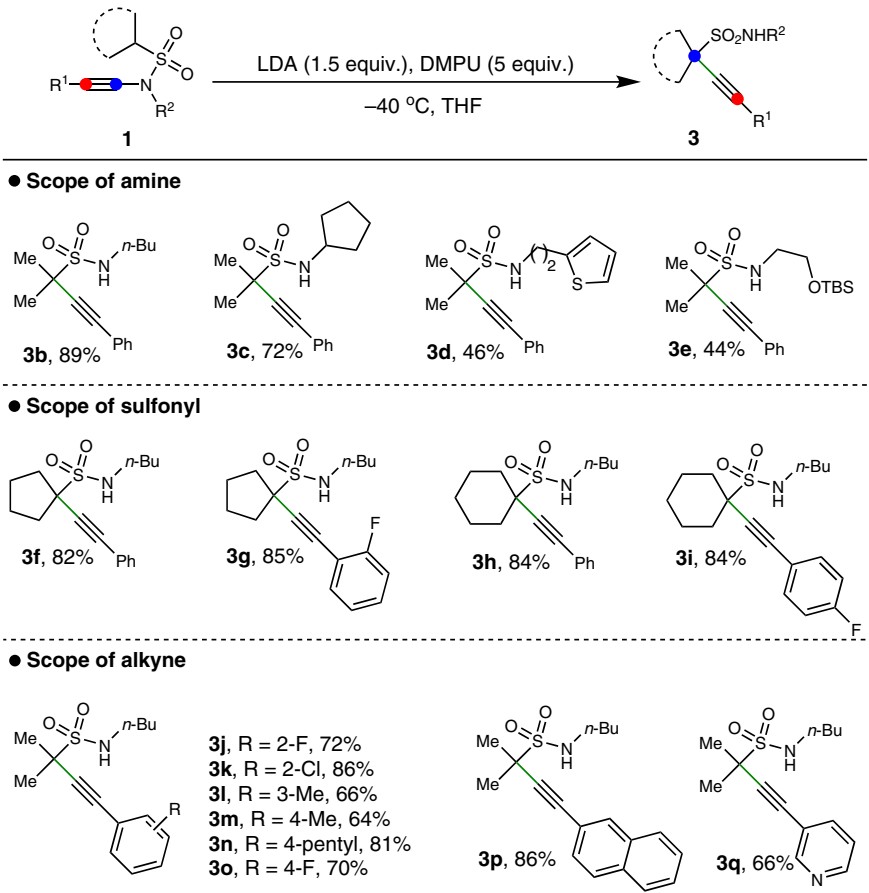

**Fig. 5 Scope of propargyl sulfonamides.** Standard condition: **1** (0.2 mmol), LDA (0.3 mmol), DMPU (1 mmol), THF (2 mL), −40 °C, 1 h, then treated with MeOH (0.1 mL). Yields refer to isolated products.

reorganization process of ynamide **1a** was quenched with MeI, an *N*-methyl propargyl sulfonamide product **5** was obtained in good yield (Fig. 6c). This experiment suggested the existence of sulfonamide anion intermediate after this skeletal reorganization process.

In order to view more insights into the reaction, the mechanism exploration was conducted. First, the crossover reaction was carried out to further explore the molecular reaction mode of this skeleton reorganization. As shown in Fig. 6d, when ynamide **1r** and **1x** were subjected in one pot under the standard reaction condition, **2q** and **2v** were exclusively formed in 79 and 73% yields, respectively. This result excluded the possibility of intermolecular group exchange and verified this reaction is an intramolecular process. Besides, treating the mixture of **1c** and MeI with LDA could exclusively deliver the α-methylation product **6** in an excellent 91% yield (Fig. 6e). This outcome disclosed that the α-lithiation was the initial step of the skeleton reorganization. Meanwhile, isotope-labelling experiment was conducted to probe the skeleton reorganization processes. As shown in Fig. 6f, (β-[13]C)-**1a** was prepared and subjected to the standard reactions. [13]C-**2a** and [13]C-**3a** could be formed smoothly under standard conditions. The position of [13]C-labelled carbon atom in the products was identified, which indicated that phenyl migration did not occur in these skeleton reorganization processes.

Moreover, we conducted density functional theory (DFT) calculations at the level of M06 using the Gaussian 09 suite of computational programs. The 6-31G(d,p) basis set was applied for the C, H, O, N, S and Li atoms. In the LDA-mediated process,

there are three possible cyclizations after α-lithiation (Fig. 7). In paths 1 and 2, Li species **Int-A** may undergo a 4-exo-dig cyclization to deliver isometric **Int-B** or **Int-C** via transition state **TS₁** or **TS₂** with a barrier of 17.6 or 18.4 kcal/mol, respectively. In path 3, **Int-A** undergoes a 5-endo-dig cyclization to give **Int-D** by **TS₃** with a higher barrier of 20.4 kcal/mol. Similarly, in the presence of DMPU, the formed Li species **Int-E** may undergo three types of cyclizations as well (Fig. 8). In paths 4 and 5, **Int-E** may transform to **Int-F** or **Int-G** via **TS₄** or **TS₅** with a barrier of 22.5 or 23.7 kcal/mol. However, in path 6, **Int-E** needs a higher barrier of 25.2 kcal/mol to initiate a 5-endo-cyclization to deliver **Int-H**. The relative lower barrier of paths 1 and 4 suggests that 4-exo-dig cyclization/*cis* addition is more favoured in these two skeleton reorganization processes.

Based on experimental and computational studies, a plausible reaction mechanism for this skeletal reorganization divergence is proposed (Fig. 9). Initially, the α-position of sulfonyl group would undergo lithiation upon treatment with LDA to deliver **A** and then a *4-exo-dig* cyclization would occur to generate four-membered β-sultam intermediate **B**[63,64,67,68]. Subsequently, **B** presumably undergoes a 1,3-sulfonyl migration to form lithium thiete sulfone intermediate **C**[69,70], which could be protonated by MeOH to deliver product **2**. When MeOH was replaced by other electrophilic reagents, intermediate **C** could be functionalized directly to obtain *N*-substituted thiete sulfone products. In the presence of DMPU, α-lithiation of **1** could deliver Li species **D**, which undergoes a 4-exo-dig cyclization to form intermediate **E**. Unlike intermediate **B**, intermediate **E** may undergo a β-elimination to generate lithium propargyl sulfonamide **F**,

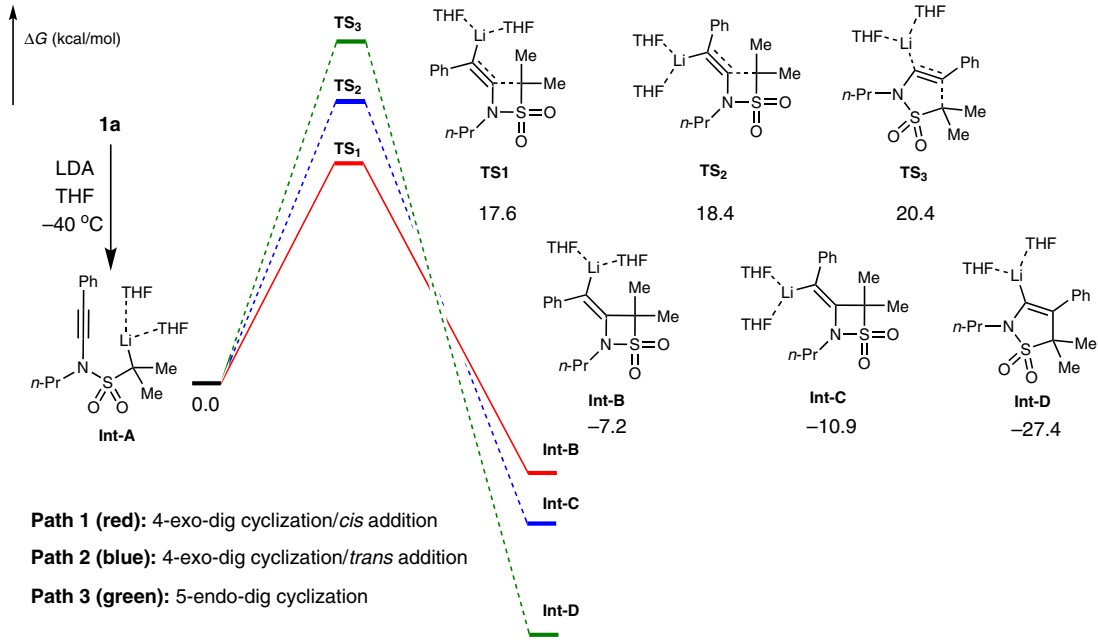

**Fig. 6 Synthetic applications and mechanistic experiments.** Gram-scale reaction of ynamide **1a**. **b** Electrophilic reagents-quenched reorganization toward *N*-functionalized thiete sulfones **4**. **c** MeI-quenched reorganization toward *N*-methyl propargyl sulfonamide **5**. **d** Crossover reaction. **e** α-Methylation experiment. **f** [13]C-Labelled experiment.

**Fig. 7 DFT calculations for the LDA-mediated cyclization in the skeleton reorganization.** Free energy profiles (kcal/mol) of three possible cyclizations from **Int-A**. path 1 (red): 4-exo-dig cyclization/*cis* addition leading to **Int-B** (via **TS₁**), path 2 (blue): 4-exo-dig cyclization/*trans* addition leading to **Int-C** (via **TS₂**), path 3 (green): 5-endo-dig cyclization leading to **Int-D** (via **TS₃**).

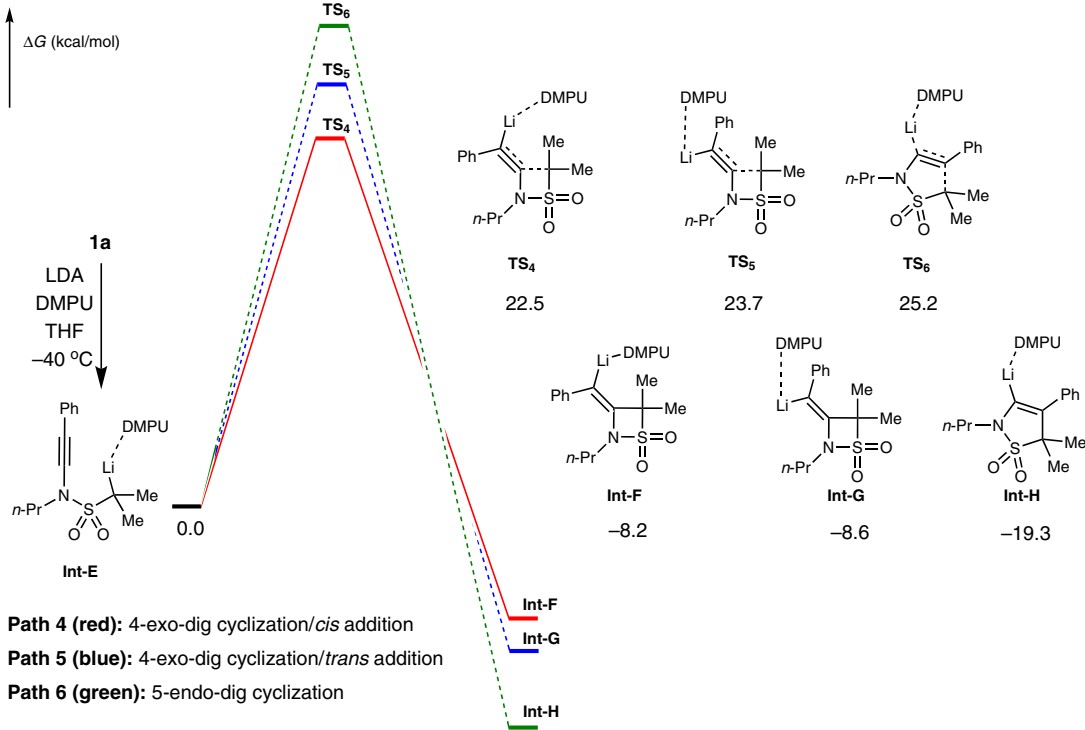

**Fig. 8 DFT calculations for the LDA/DMPU-mediated cyclization in the skeleton reorganization.** Free energy profiles (kcal/mol) of three possible cyclizations from **Int-E**. path 4 (red): 4-exo-dig cyclization/*cis* addition leading to **Int-F** (via **TS₄**), path 5 (blue): 4-exo-dig cyclization/*trans* addition leading to **Int-G** (via **TS₅**), path 6 (green): 5-endo-dig cyclization leading to **Int-H** (via **TS₆**).

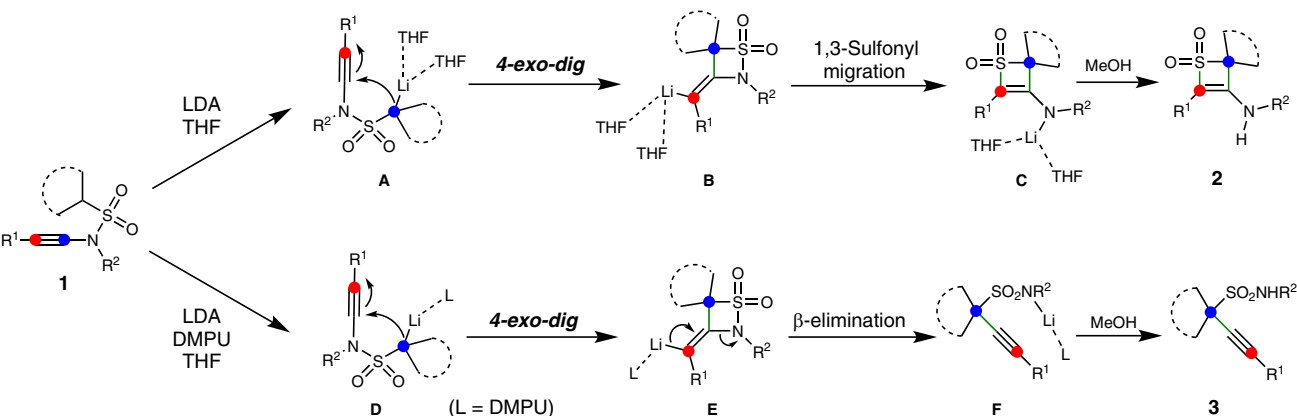

**Fig. 9 Plausible mechanism.** Possible reaction mechanism of the skeletal reorganization divergence.

probably due to the addition of DMPU that dissociates the intermolecular or intramolecular interaction of Li species and change the reactivity. The following protonation would furnish product **3**.

## Discussion

In summary, we have discovered a skeletal reorganization divergence of *N*-sulfonyl ynamides. Upon treatment with lithium base, the *N*-sulfonyl ynamides could undergo lithiation/cyclization and the sequential ligand-determining 1,3-sulfonyl migration or β-elimination to deliver thiete sulfones or propargyl sulfonamides. This skeletal reorganization divergence features broad substrate scope and scalability. Mechanistic experiments and DFT calculations are conducted to verify the rationality of the proposed mechanism. Therefore, this protocol not only represents a new skeletal reorganization mode but also provides facile and selective access to privileged molecules from the easily accessible ynamides.

## Methods

**General procedure for the synthesis of thiete sulfones 2**. An oven-dried Schlenk tube equipped with a magnetic stirrer bar was purged with argon three times. Ynamide **1** (0.2 mmol) was dissolved in 2 mL anhydrous tetrahydrofuran (THF) and added by a syringe. The mixture was cooled to −40 °C and LDA (2 mol/L in THF, 0.15 mL, 0.3 mmol) was added dropwise. The reaction was stirred at −40 °C for another 1 h. MeOH (0.1 mL) was added to quench the reaction and then the mixture was concentrated under vacuum to obtain the residue, which was further purified by silica gel column chromatography using ethyl acetate/petroleum ether as eluent to give thiete sulfones **2**.

**General procedure for the synthesis of propargyl sulfonamides 3**. An oven-dried Schlenk tube equipped with a magnetic stirrer bar was purged with argon three times. Ynamide **1** (0.2 mmol) was dissolved in 2 mL anhydrous THF and added by a syringe. DMPU (62 μL, 1 mmol) was added and the mixture was cooled to −40 °C. Subsequently, LDA (2 mol/L in THF, 0.15 mL, 0.3 mmol) was added dropwise. The reaction was stirred at −40 °C for another 1 h, and then MeOH (0.1 mL) was added to quench the reaction. The mixture was concentrated under vacuum to obtain the residue, which was further purified by silica gel column chromatography using ethyl acetate/petroleum ether as eluent to give propargyl sulfonamides **3**.

## Data availability

All data generated and analysed during this study are included in this article and its Supplementary Information and also available from the authors upon reasonable request. The X-ray crystallographic coordinates for structures reported in this article have been deposited at the Cambridge Crystallographic Data Centre (CCDC), under deposition numbers CCDC 1999861 (**2a**), 1999862 (**2b**), 1999863 (**2j**), 1999864 (**2k**), 1999865 (**2q**) and 1999866 (**2a'**). These data can be obtained free of charge via https://www.ccdc.cam.ac.uk/structures/.

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

## Acknowledgements

We are grateful for the financial support by the National Natural Science Foundation of China (21971222), Leading Talent of "Ten Thousand Plan"-National High-Level Talents Special Support Plan. We thank Dr. Zhenjun Mao (Department of Chemistry, Zhejiang University) and Jianyang Pan (Research and Service Center, College of Pharmaceutical Sciences, Zhejiang University) for performing NMR spectrometry for structure elucidation.

## Author contributions

L.Z., Y.L. and J.L. performed experiments and analysed the data. H.X. conducted the DFT calculations. S.C. conceived and directed the project and wrote the paper. H.S. was involved in the preparation of ynamides and discussion of the project. All authors discussed the results and commented on the manuscript.

## Competing interests

The authors declare no competing interests.
