## [Peer Review File · Nature Communications]

REVIEWER COMMENTS

Reviewer #1 (Remarks to the Author):

In this manuscript, Cui and co-workers report a LDA-mediated unexpected skeletal reorganization of N-sulfonyl ynamides, providing a new strategy for divergent synthesis of functionalized thiete sulfones and propargyl sulfonamides with broad substrate scope in generally moderate to good yields under mild conditions. Interestingly, this divergent skeletal reorganization of N-sulfonyl ynamides can be controlled by ligand. The mechanistic proposal is reasonable. The manuscript is well written, and the supplementary material is sufficiently complete. This is a very nice piece of work, which is of mechanistic and potential synthetic value in the ynamide chemistry. Therefore, this reviewer strongly recommends acceptance of the paper in Nature Communications after minor revision as described below:

- 1) Page 4, line 2, recent representative examples on the Brønsted acid-catalyzed cyclizations of ynamides should be cited in the references.
- 2) In Fig. 4, the compatibilities of alkyl-substituted ynamides and terminal ynamides should be demonstrated.
- 3) Page 7, line 6 and Page 10, line 4, "thiophene-3-ethyl" should be "thiophene-2-ethyl".
- 4) In Fig. 5, compound 3b, "Bu" should be "n-Bu".
- 5) Double check the mistakes in references, for example: Ref. 2, the author's names "Xie, X. N. & Zu, L. S." should be "Xie, X. & Zu, L.", other references such as Ref. 4, Ref. 17, contain the same mistakes. Ref. 12, "L. Li, Z.-L. Li, Q.-S. Gu, N. Wang, X.-Y. Liu," the format of these names is wrong. Ref. 10, "Tandem" should be "tandem", Ref. 25, "parallel" should be "Parallel", other references such as Ref. 16, Ref. 56, contain the same mistakes. Ref. 42 and 54, "Acs Catal." should be "ACS Catal.". Ref. 47 and 58, "Synlett," the comma should be deleted, Ref. 64, "Chem Rev" should be "Chem. Rev.". Ref. 49, "alpha-Imin" should be "α-imino".

Reviewer #2 (Remarks to the Author):

In this manuscript, Cui and coworkers report a set of two divergent skeletal reorganization of N-sulfonyl ynamides leading to thiete sulfones and propargylic sulfonamides. Upon deprotonation alpha to the sulfonamide moiety with LDA, the skeletal reorganization occurs, leading to thiete sulfones or propargylic sulphonamides depending on the presence of additional DMPU or not. These compounds can be further functionalized by trapping the resulting lithium amide with various electrophiles and the two processes seem to be fairly general and the yields are moderate to good.

While the two processes are fairly interesting from a mechanistic point of view and definitely represent an interesting addition to the chemistry of ynamides, the products obtained are quite specific and their synthetic usefulness is far to be obvious. Indeed, there are much simpler way to prepare propargylic sulfonamides and amino-thiete sulfones are peculiar molecules that might not be of interest to that many chemists.

As mentioned above, the main novelty of this manuscript is the unprecedented skeletal rearrangement and the reaction mechanisms associated to these. However, the mechanistic studies are far to be convincing and they only rely on really basic experiments. The crossover reaction is in fact meaningless since a crossover is highly unlikely with this kind of anionic transformations. As for the reaction attempted from N-acetyl- derivative 1o', it does not support in any way the mechanism proposed. Moreover, the authors propose a 4-exo-dig-carbolithiation without any explanation on why it would be favored over the complementary 5-endo-dig process which could actually, if followed by an alpha-elimination and a Fritsch-Buttenberg-Wiechell rearrangement, account for the formation of propargylic sulphonamides. Many other mechanistic pathways are possible and they do not even seem to have been considered by the authors. For

this manuscript to be potentially published in Nature Communications, extensive and properly conducted mechanistic investigations should be performed, notably with ^{13}C labelled substrates that would give crucial information on the skeletal rearrangement and DFT studies. The key role of DMPU should also be rationalized and not swept under the carpet.

In its current state, the manuscript cannot be published in Nature Communications, or elsewhere, and need to be extensively revised according to the comments above and the other ones listed below. It should be reviewed again to be published in Nature Communications.

Other points to be carefully addressed:

- The yields are in the moderate range in most cases and since mixtures of thiete sulfones and propargylic sulphonamides can be obtained, the ratio of these products in crude reaction mixtures should be given for all compounds.
- The manuscript should be extensively revised by a native English speaker to correct all grammatical and typographical errors. The authors should also pay attention to the scientific writing to make sure that the manuscript is much more rigorous.
- As for the Supporting Information section, IR are needed since for such small molecules, peaks of the sulphonamide and the alkyne are characteristic.

The details for response to reviewers are listed below:

Reviewer 1

(1) Page 4, line 2, recent representative examples on the Brønsted acid-catalyzed cyclizations of ynamides should be cited in the references.

Answer: A representative example has been cited as ref. 59.

(2) In Fig. 4, the compatibilities of alkyl-substituted ynamides and terminal ynamides should be demonstrated.

Answer: The alkyl-substituted ynamides and terminal ynamides were prepared and tested in this process and found not suitable in this reaction (Fig. 4). We also added a comment in text as “Unfortunately, β -alkyl substituted and terminal ynamides were not applicable in this process (2e'-2f').”.

(3) Page 7, line 6 and Page 10, line 4, “thiophene-3-ethyl” should be “thiophene-2-ethyl”.

Answer: This has been corrected.

(4) In Fig. 5, compound 3b, “Bu” should be “n-Bu”.

Answer: This has been corrected.

(5) Double check the mistakes in references, for example: Ref. 2, the author's names “Xie, X. N. & Zu, L. S.” should be “Xie, X. & Zu, L.”, other references such as Ref. 4, Ref. 17, contain the same mistakes. Ref. 12, “L. Li, Z.-L. Li, Q.-S. Gu, N. Wang, X.-Y. Liu,” the format of these names is wrong. Ref. 10, “Tandem” should be “tandem”, Ref. 25, “parallel” should be “Parallel”, other references such as Ref. 16, Ref. 56, contain the same mistakes. Ref. 42 and 54, “Acs Catal.” should be “ACS Catal.”. Ref. 47 and 58, “Synlett,” the comma should be deleted, Ref. 64, “Chem Rev” should be “Chem. Rev.”.

Ref. 49, “alpha-Imin” should be “ α -imino”.

Answer: These have been corrected.

Reviewer 2

(1) The crossover reaction is in fact meaningless since a crossover is highly unlikely with this kind of anionic transformations.

Answer: Actually, we think the crossover reaction could exclude the possibility of intermolecular group exchange of this reaction.

(2) As for the reaction attempted from N-acetyl- derivative 1o', it does not support in any way the mechanism proposed.

Answer: This has been removed.

(3) Moreover, the authors propose a 4-exo-dig-carbolithiation without any explanation on why it would be favored over the complementary 5-endo-dig process which could actually, if followed by an alpha-elimination and a Fritsch-Buttenberg-Wiechell rearrangement, account for the formation of propargylic sulphonamides. Many other mechanistic pathways are possible and they do not even seem to have been considered by the authors. For this manuscript to be potentially published in Nature Communications, extensive and properly conducted mechanistic investigations should be performed, notably with ^{13}C labelled substrates that would give crucial information on the skeletal rearrangement and DFT studies.

Answer: ^{13}C labelled substrate was prepared and subjected to the reaction. The ^{13}C NMR analysis showed that there is not phenyl migration in these two reorganizations.

For the formation of propargylic sulphonamides, the result could exclude the process of 5-endo-dig cyclization/ alpha-elimination/Fritsch-Buttenberg-Wiechell rearrangement.

In addition, DFT calculations were conducted, and the result suggested that the 4-exo-dig *syn*-carbolithiation was more favored in these two reorganizations. A plausible reaction mechanism is proposed in Fig. 8.

(4) The key role of DMPU should also be rationalized and not swept under the carpet.

Answer: We speculate that the addition of DMPU could dissociate the intermolecular interaction of intermediate Li-species, and shifts the reactivity of Li-species to furnish the skeletal reorganization divergence.

(5) The yields are in the moderate range in most cases and since mixtures of thiete sulfones and propargylic sulphonamides can be obtained, the ratio of these products in crude reaction mixtures should be given for all compounds.

Answer: Crude ¹H NMR were conducted to identify the ratio of these products (see Supporting Information).

(6) The manuscript should be extensively revised by a native English speaker to correct all grammatical and typographical errors. The authors should also pay attention to the scientific writing to make sure that the manuscript is much more rigorous.

Answer: The manuscript has been well checked.

(7) As for the Supporting Information section, IR are needed since for such small molecules, peaks of the sulphonamide and the alkyne are characteristic.

Answer: IR data for all products have been added in the Supporting Information.

REVIEWERS' COMMENTS

Reviewer #1 (Remarks to the Author):

This manuscript has been extensively revised and all comments have been carefully addressed: this manuscript is therefore now publishable in the Nature Communications.

Reviewer #2 (Remarks to the Author):

Based on the revised manuscript and the response letter, this manuscript has been carefully revised and could be published in Nature Communications.